# Thyme Essential Oil as a Potential Tool Against Common and Re-Emerging Foodborne Pathogens: Biocidal Effect on Bacterial Membrane Permeability

**DOI:** 10.3390/microorganisms13010037

**Published:** 2024-12-27

**Authors:** Martina Di Rosario, Leonardo Continisio, Giuseppe Mantova, Francesca Carraturo, Elena Scaglione, Daniela Sateriale, Giuseppina Forgione, Chiara Pagliuca, Caterina Pagliarulo, Roberta Colicchio, Mariateresa Vitiello, Paola Salvatore

**Affiliations:** 1Department of Molecular Medicine and Medical Biotechnologies, University of Naples “Federico II”, 80131 Naples, Italy; martina.dirosario@unina.it (M.D.R.); leonardo.continisio01@universitadipavia.it (L.C.); giuseppe.mantova@unina.it (G.M.); elena.scaglione@unina.it (E.S.); chiara.pagliuca@unina.it (C.P.); roberta.colicchio@unina.it (R.C.); 2PhD National Programme in One Health Approaches to Infectious Diseases and Life Science Research, Department of Public Health, Experimental and Forensic Medicine, University of Pavia, 27100 Pavia, Italy; 3Pediatric Surgery Unit, Department of Transalational Medical Science, University of Naples “Federico II”, 80131 Naples, Italy; fr.carraturo@studenti.unina.it; 4Department of Science and Technology, University of Sannio, Via F. De Sanctis snc, 82100 Benevento, Italy; sateriale@unisannio.it (D.S.); gforgione@unisannio.it (G.F.); caterina.pagliarulo@unisannio.it (C.P.); 5CEINGE-Biotecnologie Avanzate Franco Salvatore s.c.ar.l., 80131 Naples, Italy; 6Task Force on Microbiome Studies, University of Naples “Federico II”, 80131 Naples, Italy

**Keywords:** thyme essential oil, antibacterial effect, foodborne pathogens, bacterial membrane damage

## Abstract

Over the past decade, foodborne diseases have become a significant public health concern, affecting millions of people globally. Major pathogens like *Salmonella* spp., *Escherichia coli*, *Listeria monocytogenes*, and *Staphylococcus aureus* contaminate food and cause several infections. This study investigates the potential of thyme essential oil (Thy-EO) as a natural antimicrobial agent against most common and re-emerging foodborne bacteria, including *S. enterica*, *Yersinia enterocolitica*, and *L. monocytogenes*. Preliminary tests provided qualitative evidence of Thy-EO’s efficacy by evaluating its antibacterial activity through direct contact and vapor phase exposure. Then, the Minimum Inhibitory Concentration (MIC) and Minimum Bactericidal Concentration (MBC) were defined to quantitatively evaluate the bacteriostatic and bactericidal effects of Thy-EO, revealing a strong inhibitory effect against *S. enterica*, *Y. enterocolitica* and *L. monocytogenes*. Additionally, Thy-EO exerted rapid bactericidal kinetics characterized by the disruption of bacterial cell membrane integrity over time. Results highlight Thy-EO’s potential as an alternative antimicrobial agent, demonstrating that treatment with Thy-EO significantly and irreversibly affects the growth of the tested foodborne pathogens.

## 1. Introduction

In the last decade, diseases caused by foodborne pathogens have increased compared to the past and have become one of the most widespread public health problems in the world, affecting millions of people each year [1,2]. These illnesses result from the consumption of contaminated and perishable foods, and they can manifest in various forms, ranging from mild gastrointestinal discomfort to severe and life-threatening conditions. It is well known that many stress-resistant bacteria can contaminate food products [3] and among these *Salmonella* spp., enterohemorrhagic *Escherichia coli*, *Listeria monocytogenes*, *Bacillus* spp., *Staphylococcus aureus*, and *Clostridium* spp. are of great significance [3,4].

Emerging or re-emerging foodborne pathogens result from a combination of factors that facilitate their spread, including pathogen characteristics, environmental conditions, food production, distribution practices, and consumer behavior [5]. The increasing prevalence of these pathogens requires a greater understanding of these organisms, contributing to more effective control measures. In addition to the emergence of foodborne pathogens, toxins, and resistance mechanisms represent significant challenges to food safety, complicating the diagnosis and treatment of outbreaks [6,7].

Food safety and control systems must adapt to industry-wide manufacturing and distribution practices, gradually shifting the focus from the final product to process control throughout the food chain. Therefore, food safety must be systematically integrated into policies and interventions to improve nutrition and food security.

The increased incidence of foodborne threats combined with new social and economic implications leads to an urgent need for safer food through the development of new and nontoxic preservative agents with antimicrobial and antioxidant properties. In the last two decades, much attention has been focused on food bio-preservation, a “green strategy” that can assure shelf-life extension and food safety [8,9].

Therefore, thanks to the great interest of the food industries to find more sustainable alternative solutions to synthetic preservatives, and to ensure the safety of the product distributed on the market, we have focused on the research of new natural antimicrobial compounds [10,11,12]. From antiquity, herbs and spices have been used for food, flavorings, medicine and cooking, and food preservatives in recent decades [13]. Many spices—such as cloves, oregano, thyme, cumin, and caraway—are used to treat infectious diseases or as food preserving agents, due to their experimental antimicrobial activity against pathogenic and spoilage fungi [14]. In addition, it is known that secondary metabolites of these spices have antimicrobial activity, most of which are generally recognized as safe substances for use in food, with negligible adverse effects [15,16].

Considering the increasing interest of consumers towards natural products, essential oils (EOs) may constitute effective alternatives or complements to synthetic compounds, without showing resistance effects [17]. However, it is known that the composition of natural products, including EOs, is importantly affected by many factors such as environmental variation. Moreover, with the advances of new technologies, as well as computational and analytical methods, it is becoming increasingly feasible to isolate compounds with more reproducible therapeutic effects from natural extracts [18].

EOs are secondary metabolites derived from plants of the following families: *Asteraceae*, *Lamiaceae*, *Lauraceae*, *Myrtaceae*, *Rutaceae*, *Umbelliferae*, *Zingiberaceae*, and others and consist of a complex mixture of low molecular mass volatile compounds [19]. The beneficial properties of the EOs are mainly attributed to the presence of bioactive compounds (BACs), responsible for their anticarcinogenic, antidiabetic, anti-inflammatory, antimutagenic, antioxidant, and antiproliferative properties [20].

The antimicrobial properties of EO extracts from many plants such as basil, citrus, fennel, lemongrass, oregano, rosemary, and thyme are well-known for their ability to synthesize aromatic substances such as phenols or oxygen-substituted derivatives [21].

BACs present different structural groups in their composition that may act on different targets of bacterial cells. Eos’ activity is related to bacteriostatic or bactericidal activity and the disruption of the membrane is the most common mechanism [22]. The hydrophobicity/lipophilic nature of EOs allows them to accumulate in the phospholipid bilayer of the cytoplasmic membrane, resulting in a loss of membrane structure, impermeability, cellular components, and functions [22]. Recent studies have demonstrated their antibiofilm, antifungal, antiviral, and anticancer properties [11,23,24,25]. Due to their antimicrobial properties, EOs today provide an important natural alternative to antimicrobial agents.

Over the years, the antimicrobial and antioxidant potential of the *Thymus* species EOs (Thy-EOs), associated with the presence of pharmacologically active constituents (higher contents of phenolic compounds such as thymol and carvacrol), has been documented to confirm their potential industrial and health applications [26,27,28]. Thymol, a major constituent of *T. vulgaris* EOs, has been extensively studied for its multiple therapeutic properties, highlighting its anti-inflammatory, antiproliferative, antifungal, and antibacterial effects [27].

We have previously demonstrated that Thy-EOs present an excellent antibacterial and antibiofilm activity against significant food pathogens, *S. enterica* subsp. *enterica* serovar Typhimurium and *B. cereus*, suggesting their use as a potential natural food preservative to counteract biofilm-related infections in the food industry [11].

In this context, the purpose of this study was to extend the functional potential of Thy-EO by assessing its antibacterial activity against three major foodborne bacteria like *S. enterica* group B, *Y. enterocolitica*, and *L. monocytogenes* and to investigate the possible underlying inhibitory mechanism. In vitro tests were performed to evaluate the antimicrobial effects of Thy-EO on the selected pathogens both by direct contact and in the vapor phase. The bactericidal kinetics as well as the disruption of bacterial cell membrane integrity over time were evaluated. The results may contribute to the development of alternative, novel preventive and/or antibiotic independent therapeutic measures that could improve food safety and reduce reliance on synthetic preservatives.

## 2. Materials and Methods

### 2.1. Reagents and Chemicals

Thyme essential oil (thyme oil- white FG, W306509, Sigma Aldrich, St. Louis, MO, USA) (Thy-EO), stored in the dark at 25 °C until further use, was purchased from Merck (Darmstadt, Germany) in order to ensure uniformity in its chemical compositions, avoiding possible alterations caused by climatic and geographic factors. Its chemical analysis was reported previously and was characterized by a total of 15 components, among these the major was thymol (43.2%) followed by p-cymene (321.65%), linalool (5.38%), and carvacrol (5.11%) [29]. Thy-EO was chosen due to its broad-spectrum antimicrobial activity, even at low concentrations, and a pleasant aroma. Tryptic Soya (TS) Broth/Agar and Muller Hinton (MH) agar (Oxoid S.p.a., Milan, Italy), were used as bacterial culture media. Glycerol (Carlo Erba, Reagents, Milan, Italy) was employed for the storage of bacterial isolates at −80 °C prior to their utilization. In addition, the antibiotic Gentamicin (Merck KGaA, Readington, NJ, USA), NaClO 4.5% (Selex, Trezzano Sul Naviglio, Milan, Italy), and Triton X-100 1% (Medical Chemical Corporation, Torrance, CA, USA) were utilized in subsequent experiments.

### 2.2. Clinical Bacterial Strains and Growth Conditions

The antimicrobial activity of Thy-EO was evaluated against three foodborne bacterial pathogens including *S. enterica* group B, *Y. enterocolitica*, and *L. monocytogenes.* The bacterial clinical isolates were identified by mass spectrometry using a matrix-assisted laser desorption/ionization (MALDI) mass spectrometer (Bruker Daltonics, MALDI Biotyper, Fremont, CA, USA). To investigate the antimicrobial susceptibility to *S. enterica* group B, *Y. enterocolitica*, and *L. monocytogenes*, we performed antimicrobial susceptibility testing (AST) with the VITEK^®^ 2 system, according to the manufacturer’s instructions, using the software version. Bacterial growths were performed in aerobic conditions at 37 °C in a TS Broth/Agar and in MH agar. The isolates were stored frozen at −80 °C in TS broth supplemented with 10% glycerol (*v*/*v*) until use and the working cultures were activated in agar or broth medium at 37 °C for 15–18 h.

### 2.3. Agar Diffusion Assay

In vitro antimicrobial activity of Thy-EO was evaluated through the agar diffusion method. The test was performed according to the standard method [11,12,30]. Bacterial suspensions were prepared at an optical density (OD) of 0.5 OD_600nm_ and were spread onto MH agar using a sterile swab. Subsequently, paper disks were impregnated with different concentrations of Thy-EO, ranging from 5 to 15 µL. Distilled water and Gentamicin (100 μg/mL) were used as negative and positive controls, respectively. The cultures were then incubated overnight at 37 °C. The inhibitory effect of Thy-EO was evaluated by measuring the diameter of inhibition zone around disk [12]. All experiments were conducted in triplicate, using three independent bacterial cultures. The antibacterial activity of Thy-EO was quantified by measuring the mean diameter of the inhibition zone (MDIZ) in mm [11,12,30]. The data were analyzed and presented graphically using “GraphPad Prism 6” software. Results are expressed as the mean ± Standard Deviation (SD). Statistical significance was evaluated using Student’s *t*-test (*p* value ≤ 0.05).

### 2.4. Broth Dilution Method

Thy-EO’s quantitative antimicrobial activity was assessed against *S. enterica*, *Y. enterocolitica*, and *L. monocytogenes* through the broth dilution method. In brief, bacteria were grown in TS broth at 37 °C until the late-log phase. Then, bacterial suspensions were diluted in Phosphate-Buffered Saline (PBS 1X), and an inoculum of 1 × 10^5^ colony forming units/mL (CFU/mL) of each tested isolate was used, according to the Clinical and Laboratory Standards Institute [31]. Increasing concentrations of Thy-EO (0.125, 0.2, 0.3, 0.4, 0.5, 0.62, 1.25, 2.5 μL/mL) were tested against each foodborne bacterial pathogen. The samples were then incubated overnight at 37 °C with constant shaking to maintain homogeneous aggregates of Thy-EO micelles in the broth medium [11]. After the incubation time, the Minimum Inhibitory Concentration (MIC) was obtained by measuring the reduction of OD_600nm_. To determine the Minimum Bactericidal Concentration (MBC), dilutions of each bacterial suspension were plated on TS agar and the number of viable bacteria was evaluated by the CFU method. MIC values were associated with the lowest concentration of Thy-EO that prevented microbial growth, while MBC values were defined as the minimum concentration of Thy-EO that killed the bacteria in the initial inoculum. All assays were performed in triplicate with three independent cultures.

### 2.5. Disk-Volatilization Assay

The disk-volatilization method was employed to assess the antimicrobial efficacy of Thy-EO in the vapor phase against *S. enterica*, *Y. enterocolitica*, and *L. monocytogenes* [32]. Each bacterial strain was resuspended in MH broth at 0.500 OD_600nm_ and then spread on the surface of a MH agar medium using a sterile swab. Subsequently, paper disks were placed on the inside of the Petri plate lid and different concentrations of Thy-EO were added to each disk (5, 10, 15 μL). Distilled water and NaClO 4.5% were used as negative and positive control, respectively. Plates inoculated with each bacterial strain were immediately inverted onto the lid and sealed with parafilm to prevent the leakage of Thy-EO volatile compounds, then incubated at 37 °C for 24 h. All experiments were performed in triplicate with three independent cultures, and the antibacterial activity of Thy-EO volatile compounds was expressed as the mean diameter of the inhibition zone (MDIZ) in mm. The data were analyzed and presented graphically using GraphPad Prism 6 software. Results are shown as mean ± SD, and statistical significance was determined using Student’s *t*-test (*p* value ≤ 0.05).

### 2.6. Time-Kill Kinetics Assay

Time-kill assays were performed to evaluate the effect of Thy-EO on selected foodborne bacterial clinical isolates. Bacteria were grown in TS broth at 37 °C up to the late logarithmic phase. At the end of the incubation time, the bacterial suspensions were diluted 1:100 in PBS 1X a concentration of approximately 10^6^ CFU/mL and then incubated at 37 °C with or without Thy-EO at concentrations corresponding to MIC, 2×MIC and 4×MIC value. After 1, 2, 3, 6, and 24 h of incubation, 10-fold serial dilutions of each reaction were spread on TS agar. The plates were then incubated for 24 h at 37 °C and viable counts were evaluated by the CFU method. All experiments were carried out in triplicate with three independent cultures. Results were analyzed and presented graphically using GraphPad Prism 6 software. Data were reported as mean ± SD. Statistical significance was determined by Student’s *t*-test (*p* value ≤ 0.05).

### 2.7. Membrane Permeability Assay

EOs rich in volatile monoterpenes such as thymol and carvacrol typically exhibit antimicrobial effect due to the disruption of the lipid fraction of the plasma membrane, resulting in the leakage of intracellular materials [22,33]. To evaluate the disruption of cell membrane integrity induced by the Thy-EO on the tested foodborne bacteria, the membrane permeability assay was conducted to quantify the leakage of double-strand DNA (dsDNA) and protein. Specifically, 1% of the overnight culture was added to fresh TS broth and cultured at 37 °C for 1 h until an OD_600nm_ of 0.12 to 0.14 was reached. Then, increasing concentrations of Thy-EO equal to MIC, 2×MIC, and 4×MIC value were incubated with the microbial suspension at 37 °C, with shaking at 200 rpm for 2 h. Triton X-100 1% and TS broth were used as a positive and negative control, respectively. At regular time intervals (0, 30, 60, 90, 120 min) the samples were centrifugated at 3000 rpm for 10 min (Eppendorf, Hamburg, Germany), and the cell-free supernatants were read using a UV-Visible NanoDrop™ Spectrophotometer (Thermo Fisher Scientific, Waltham, MA, USA) at 260 and 280 nm to quantitatively determine the dsDNA and protein leakage, respectively. The quantity of released dsDNA was expressed in ng/μL, while the amount of released protein was expressed in mg/mL. Three replicates were performed with three independent cultures and the results were analyzed and presented graphically using GraphPad Prism 6 software. Data were reported as mean ± SD. Statistical significance was determined by Student’s *t*-test (*p* value ≤ 0.05).

## 3. Results

### 3.1. In Vitro Antibacterial Activity of Thy-EO

The phenotypic characterization of the clinical strains was conducted to determine their antibiotic susceptibility profiles. Susceptibility tests were performed on selected bacterial isolates using antibiotics commonly employed in human therapy. The antimicrobial susceptibility test revealed that *S. enterica* group B was resistant to Quinolone antibiotics, while susceptibility to several classes of antibiotics was demonstrated (Appendix A). AST analyses for *Y. enterocolitica* showed a resistance profile to Ampicillin antibiotic with a MIC of 256 μg/mL and to Benzylpenicillin antibiotic with a MIC of 10 μg/mL (Appendix A). While against *L. monocytogenes*, AST analyses showed that the bacteria were resistant to Piperacillin antibiotic with a MIC of 3 μg/mL (Appendix A).

Antimicrobial activity tests revealed that Thy-EO exhibited a strong inhibitory action against *S. enterica*, *Y. enterocolitica*, and *L. monocytogenes*, as demonstrated by the evaluation of the inhibition zone generated by each concentration of oil tested with the agar diffusion method. The antimicrobial activity of Thy-EO was expressed as MDIZ (Figure 1). In detail, for *S. enterica*, the MDIZ ranged from 15 ± 1 mm to 18.86 ± 1.5 mm for treatment with 5 µL/disk and 10 µL/disk of Thy-EO, while a MDIZ value of 19 ± 1 was obtained with a Thy-EO concentration corresponding to 15 µL/disk (Figure 1A).

For *Y. enterocolitica*, the MDIZ values were 23 ± 2.6 mm, 25.6 ± 3.2 mm, and 27.3 ± 2 mm after treatment with a concentration of 5 µL/disk, 10 µL/disk, and 15 µL/disk of Thy-EO, respectively. The higher activity of Thy-EO, by using volumes corresponding to 5 µL/disk, 10 µL/disk, and 15 µL/disk, was demonstrated against *L. monocytogenes* isolate, with MDIZ values of 20 ± 4.5 mm, 25.6 ± 0.5, and 29 ± 1 mm, respectively. Gentamicin (100 µg/mL) was used as positive control (Figure 1). No effects were observed with the negative control.

The antimicrobial activity of Thy-EO against foodborne clinical isolates was quantitatively assessed using the broth dilution method, following the CLSI guidelines [31]. Thy-EO exerted a strong antibacterial effect against all foodborne clinical isolates tested. In detail, for *S. enterica* the MIC value was 0.4 µL/mL and the MBC value ranged from 0.4 to 0.5 µL/mL (Table 1). For *Y. enterocolitica*, the MIC value was 0.2 µL/mL while the MBC value ranged from 0.4 to 0.5 µL/mL (Table 1). Finally, a strong efficacy was observed against *L. monocytogenes*, as indicated by a MIC value of 0.125 µL/mL and MBC values ranging from 0.2 to 0.3 µL/mL (Table 1).

### 3.2. Evaluation of Antimicrobial Activity of Thy-EO Volatile Compounds

The antimicrobial properties of Thy-EO were evaluated in the vapor phase using a disk volatilization assay performed in Petri dishes. Specifically, the test was conducted using the same concentrations as those employed in the agar diffusion method, to preliminarily assess the antimicrobial activity of the compound under investigation. Similarly to the liquid phase, the MDIZ for Thy-EO vapors increased with the increasing volumes tested. The results indicated a high level of efficacy of the vapor phase against all bacterial strains evaluated. For *S. enterica*, the MDIZ value was 10.3 ± 0.5 mm after exposure to the lower Thy-EO concentration of 5 µL/disk, while the treatment with concentrations of 10 µL/disk and 15 µL/disk showed MDIZ values corresponding to 16 ± 0 and 16 ± 0.5, respectively (Figure 2A). For *Y. enterocolitica*, the MDIZ values were 16.3 ± 0.5 mm, 17.6 ± 0.5 mm, and 18 ± 0 mm after treatment with increasing concentrations of Thy-EO corresponding to 5 µL/disk, 10 µL/disk, and 15 µL/disk, respectively (Figure 2B). Similarly, Thy-EO showed a bactericidal effect against *L. monocytogenes* at all concentrations tested, equal to 5 µL/disk, 10 µL/disk, and 15 µL/disk with MDIZ values corresponding to 12.3 ± 0.5 mm, 17.6 ± 0.5, and 18.3 ± 0.6 mm, respectively (Figure 2C). NaClO 4.5%, was used as a positive control (Figure 2).

### 3.3. Evaluation of Antimicrobial Activity of Thy-EO by Time-Kill Kinetics Assay

To assess the fitness costs of Thy-EO, the growth and stationary phase survival of foodborne pathogens were monitored over 24 h, with increasing concentrations of essential oil corresponding to MIC, 2×MIC, and 4×MIC. As shown, after 1 h of incubation with Thy-EO, *S. enterica* showed an overall growth reduction throughout the observation time at Thy-EO concentrations ranging from 0.4 to 1.6 μL/mL (Figure 3A). A bactericidal effect of Thy-EO was observed against *Y. enterocolitica* after 1 h of exposure to a concentration of 0.8 μL/mL (Figure 3B). In contrast, lower concentrations, ranging from 0.2 to 0.4 μL/mL, achieved the same bactericidal effect after 2 h of exposure (Figure 3B). In addition, Thy-EO exhibited significant bactericidal activity against *L. monocytogenes*, at a concentration of 0.5 μL/mL, during the observation period. Lower concentrations of Thy-EO, at 0.125 and 0.25 μL/mL, were effective after 3 h of treatment and maintained their bactericidal effect for up to 24 h (Figure 3C). Overall, the results demonstrated that Thy-EO exerted a rapid bactericidal effect, achieving complete bacterial breakdown within 2 h. This effect remained constant and sustained, with no evidence of bacterial recovery.

### 3.4. Evaluation of the Effect of Thy-EO on Bacterial Cell Membrane Integrity

The disruption of the bacterial membrane’s integrity induced by Thy-EO was demonstrated by the leakage of genetic and protein materials through the cell membrane. The alteration of bacterial cell membrane integrity was evaluated by measuring the release of cellular components through the assessment of the absorbance at 260 and 280 nm in the supernatant of bacterial cultures treated with ranging concentrations of Thy-EO. The results demonstrated that the release of both dsDNA and protein content increased in a dose-dependent manner in bacteria exposed to increasing concentrations of Thy-EO (MIC, 2×MIC, and 4×MIC) (Figure 4). Notably, a greater effect was observed against *S. enterica*, with a dsDNA release ranging from 61 ± 9.7 to 175 ± 18.2 ng/µL in a dose-dependent manner, with the amount remaining approximately constant up to 120 min (Figure 4A). Similarly, an increase in protein release, ranging from 2.4 ± 0.1 to 7.3 ± 0.9 mg/mL, was demonstrated to be induced by Thy-EO in a dose-dependent manner throughout the observation period (Figure 4B).

Likewise, for *Y. enterocolitica*, the amount of dsDNA released ranged between 30.5 ± 8.3 and 85.2 ± 1.6 ng/µL (Figure 4C), while the extent of protein leakage ranged from 2 ± 0.5 to 3.9 ± 1.3 mg/mL (Figure 4D) in response to increasing concentrations of Thy-EO throughout the treatment period. The effects of increasing doses of Thy-EO on *L. monocytogenes* membrane integrity were found to be slightly lower. Concentrations of dsDNA released were observed to range from 4.7 ± 2.3 to 60 ± 4.3 ng/µL (Figure 4E), and the release of protein ranged from 1.1 ± 0.4 to 2.8 ± 0.6 mg/mL from 30 to 120 min of exposure (Figure 4F). Triton X-100 (1%) solution was used as positive control and confirmed the release of dsDNA and protein content through the bacterial membrane (Figure 4).

## 4. Discussion

It is known that the EOs of plant species represent a resource of compounds with broad-spectrum antimicrobial effects and significant activity against a variety of microorganisms, including bacteria, fungi, and viruses [21,25,34,35]. Over the past two decades, there has been a notable increase in scientific interest on the antimicrobial efficacy of EOs, with several studies highlighting their therapeutic potential. Due to their complex compositions, their antimicrobial effects cannot be attributable to a single mechanism; rather, it results from action on different cellular targets. This characteristic prevents the development of bacterial resistance, making them effective against multidrug-resistant bacteria [17,36,37]. Consequently, the use of EOs to inhibit the growth of pathogenic microorganisms could offer a viable and cost-effective alternative to chemical preservatives in the food industry for controlling foodborne pathogens.

Currently, the antimicrobial activity of Thy-EOs has been extensively studied, even if there are few studies regarding their efficacy against foodborne pathogens. The research conducted by Rota et al. (2008) and Diniz et al. (2023) [38,39] provided experimental evidence of the oil’s effectiveness against several bacterial strains, thereby suggesting potential applications in therapeutic and preservative contexts. Recent studies have highlighted the significant antibacterial activity of Thy-EO against foodborne pathogens, including *E. coli*, *S. aureus*, *S. enterica* subsp. *enterica* serovar Typhimurium, and *B. cereus*. This activity is attributed to the high concentrations of active phenolic compounds, such as thymol and carvacrol [8,11,27]. Due to its hydrophobic nature, thymol predominantly targets the lipid bilayer of the bacterial plasma membrane, leading to alterations in membrane permeability and subsequent leakage of cellular contents, including nucleic acids and proteins [40]. In addition, it has been demonstrated that thymol’s disruption of the cytoplasmic membrane enables its binding to bacterial DNA, resulting in structural alterations that impair gene expression and protein synthesis, thereby exerting a microbicidal effect [41,42,43].

In accordance with the current state of research, the results of our in vitro microbiological assays demonstrated that Thy-EOs exhibited strong antimicrobial activity against *S. enterica* group *B*, *Y. enterocolitica*, and *L. monocytogenes*. These emerging pathogens are widely recognized as responsible for foodborne illness outbreaks due to the contamination of food, mostly of animal origin, such as eggs and meat products [40]. Furthermore, psychotropic bacteria such as *Y. enterocolitica* and *L. monocytogenes* can proliferate at refrigeration temperatures, posing a considerable risk to food safety [41].

The evaluation of MDIZ, MIC, and MBC values as well as the growth curve of selected bacteria are used to assess the antibacterial activity of Thy-EOs. As demonstrated by the MDIZ, obtained through the agar diffusion assay, Thy-EOs exerted a more pronounced inhibitory effect against *Y. enterocolitica* at the lowest concentration, while the overall higher activity was observed against *L. monocytogenes* (Figure 1). The results of MIC analysis and growth curve indicated the same conclusion. Antibacterial screening demonstrated that Thy-EO exhibited a strong and consistent inhibitory effect against the tested foodborne pathogens, revealing MIC values between 0.125 and 0.4 μL/mL and MBC values ranging from 0.2 and 0.5 μL/mL with a remarkable activity against *L. monocytogenes* (Table 1). The different susceptibility to Thy-EOs is based on the different morphological structure of *L. monocytogenes*, that possess a monopeptide layer structure, compared to other tested Gram-negative bacteria characterized by a thick outer lipopolysaccharide layer, which provides an additional protection against antimicrobial compounds [25,44,45]. Furthermore, the low MIC and MBC values observed represent significant findings, indicating that high concentrations may not be necessary for in vivo assays based on the in vitro results. This suggests the potential for using lower doses of Thy-EO in practical applications, which may help to minimize costs and reduce the risk of adverse effects associated with higher concentrations, including possible impacts on the organoleptic (smell, taste or texture) properties of food. The analysis of Thy-EO’s volatile compounds produced similar results to those observed with the MDIZ, confirming stronger antimicrobial activity of the lowest oil volume against *Y. enterocolitica* (Figure 2).

The kinetics of bactericidal action of Thy-EOs highlighted that microbial death occurred in a time- and dose-dependent manner following exposure to the formulation. Results from the time-kill kinetic assay showed that Thy-EO exposure had a significant impact on the growth of the tested bacterial pathogens. At subinhibitory concentrations, Thy-EOs completely inhibited growth with a bactericidal effect on *S. enterica*, *Y. enterocolitica*, and *L. monocytogenes* at 1 and 3 h exposure, respectively, confirming their bactericidal activity at MIC levels (Figure 3).

Previous studies revealed similar findings, demonstrating that other essential oils also exerted inhibitory effects against various foodborne pathogenic bacteria [46,47,48].

In our investigation of the antibacterial mechanisms of Thy-EOs, we have revealed that these compounds can interact with the bacterial cell membrane. They enhance the permeability of the membrane and interfere with cellular processes, leading to a disruption of essential structural functions in bacteria and ultimately resulting in cell death at specific concentrations.

The bacterial cell membrane represents a selectively permeable barrier, regulating the exchange of substances between the inside and outside of the cell; this ensures an internal environment suitable for essential vital processes. As a result, the integrity of the cytoplasmic membrane is crucial for bacterial growth, and even small perturbations can negatively impact cellular metabolism, including enzyme activity. Therefore, examining the integrity of the cell membrane can provide deeper insights into the antibacterial action mechanism. Several researchers have reported that the measurement of specific cell leakage markers, such as absorbance at 260 nm for nucleic acids and protein content released in the bacterial supernatant, serves as a key indicator of membrane integrity, suggesting membrane sensitivity to a particular antimicrobial agent when compared to unexposed cells [45,49,50]. Our findings revealed that Thy-EOs disrupted bacterial cell membrane integrity, causing the leakage of dsDNA and protein materials. This leakage could impair the transmission of genetic information and ultimately lead to bacterial cell death. Specifically, the release of genetic and protein materials increased in a dose-dependent manner, with greater effects achieved against *S. enterica* and *Y. enterocolitica*, while a slightly lower effect was observed against *L. monocytogenes*. These findings may reflect differences on the lipid composition and net surface charge of the bacterial membranes, which could influence susceptibility to Thy-EOs’ action [28,44]. The results obtained collectively demonstrate that Thy-EO treatment caused a significant and irreversible effect on foodborne pathogen growth and cell membrane permeability, thus suggesting that the antimicrobial activity may be, at least in part, attributed to their ability to penetrate lipid assemblies, subsequently disrupting the lipid fraction of plasma membranes. This disruption leads to a significant reduction in cellular dsDNA and protein content by compromising membrane integrity. Despite the results obtained, demonstrating the potent antimicrobial effects of Thy-EOs against foodborne pathogens and positioning them as a promising candidate as a novel green preservative to prevent food contamination, several potential limitations, due to their chemical variability, volatility, lower water solubility, and allergenicity, are still questioned. A deeper understanding of how phytogenic compounds affect the gut microbiota, gut physiology, and immune system, and the underlying mechanisms within the gastrointestinal ecosystem, could provide the basis to establish their practical applicability. Moreover, further studies are required to elucidate the molecular mechanisms underlying the antibacterial effects of Thy-EOs against foodborne pathogens, and to support their potential as alternative agents in health and food safety applications.

## 5. Conclusions

The obtained results highlight the promising potential of Thy-EO as an effective antimicrobial agent against foodborne pathogens, with a focus on *S. enterica*, *Y. enterocolitica*, and *L. monocytogenes*. Its in vitro antimicrobial efficacy, characterized by low MIC and MBC values, suggests Thy-EO’s ability to inhibit and kill these pathogens at lower concentrations, minimizing potential adverse effects. The composition of Thy-EO contributes to its multiple mechanism of action, which primarily targets bacterial cell membranes and increases permeability, leading to cell disruption and death. This activity not only prevents the occurrence of resistance, often seen with conventional antibiotics, but also identifies Thy-EO as a viable alternative to synthetic preservatives in the food industry, particularly in the control of foodborne illness. This study contributes to the growing evidence supporting the use of plant compounds to fight emerging foodborne pathogens, thereby improving food safety and public health.

## Figures and Tables

**Figure 1 microorganisms-13-00037-f001:**
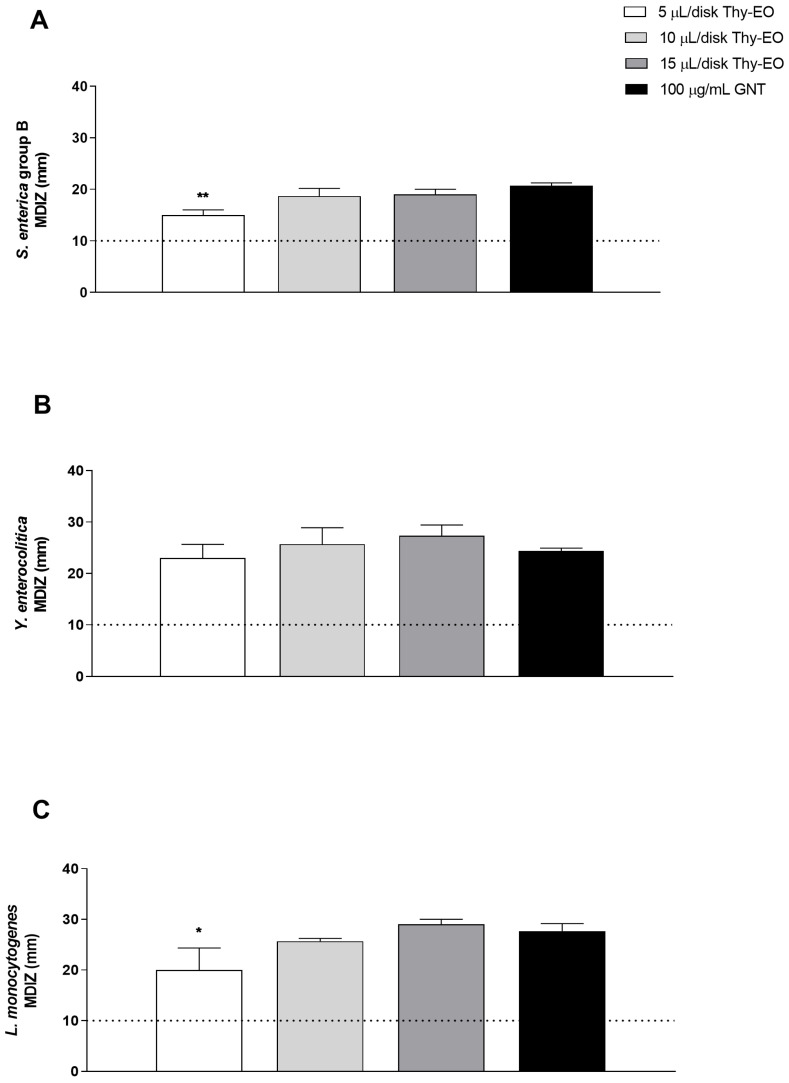
In vitro antibacterial activity of Thy-EO by the agar diffusion method. Results of the agar diffusion assay for Thy-EO against S. enterica group B (**A**), Y. enterocolitica (**B**), and L. monocytogenes (**C**) are presented. The graphical representation shows the mean diameter of the MDIZ in mm, expressed as the mean ± SD of values from triplicate assays. Gentamicin (GNT) was used as a positive control. Statistical significance was determined by Student’s *t*-test, with asterisks indicating significant differences compared to the positive control (** *p* < 0.01; * *p* < 0.05).

**Figure 2 microorganisms-13-00037-f002:**
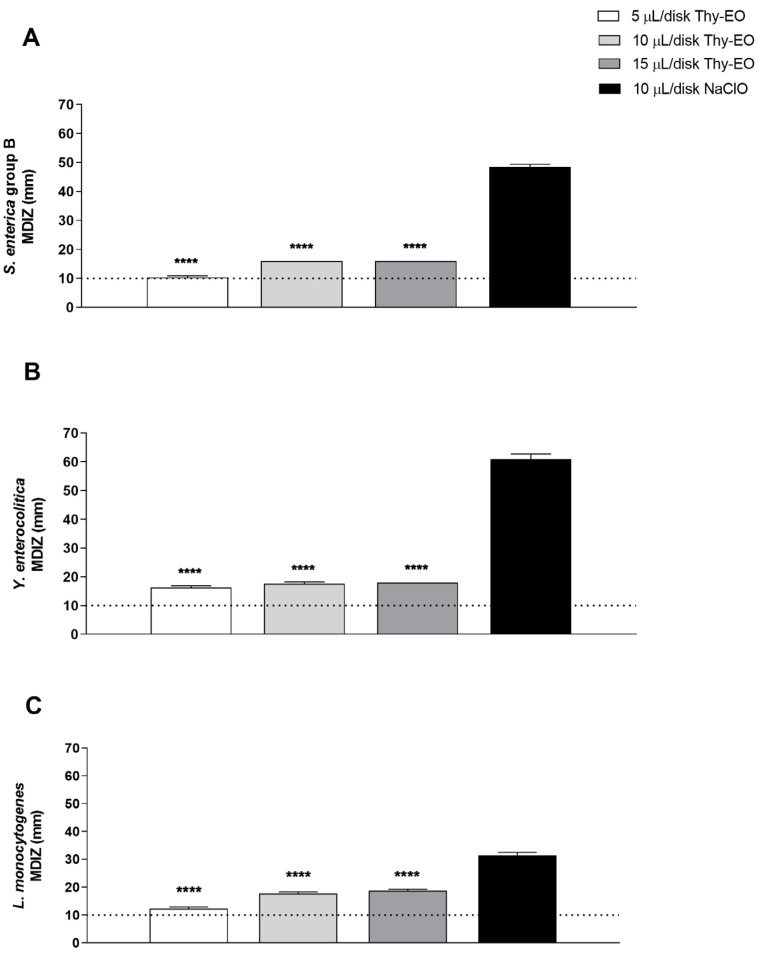
Antimicrobial effect of Thy-EO volatile compounds determined through the disk volatilization method. Evaluation of the antimicrobial activity of Thy-EO in the vapor phase against *S. enterica* (**A**), *Y. enterocolitica* (**B**), and *L. monocytogenes* (**C**). NaClO 4.5% was used as positive control. MDIZ in mm is reported as the mean ± SD of values obtained from three independent experiments. Statistical significance was determined by Student’s *t*-test, and asterisks indicate statistically significant values (**** *p* < 0.0001).

**Figure 3 microorganisms-13-00037-f003:**
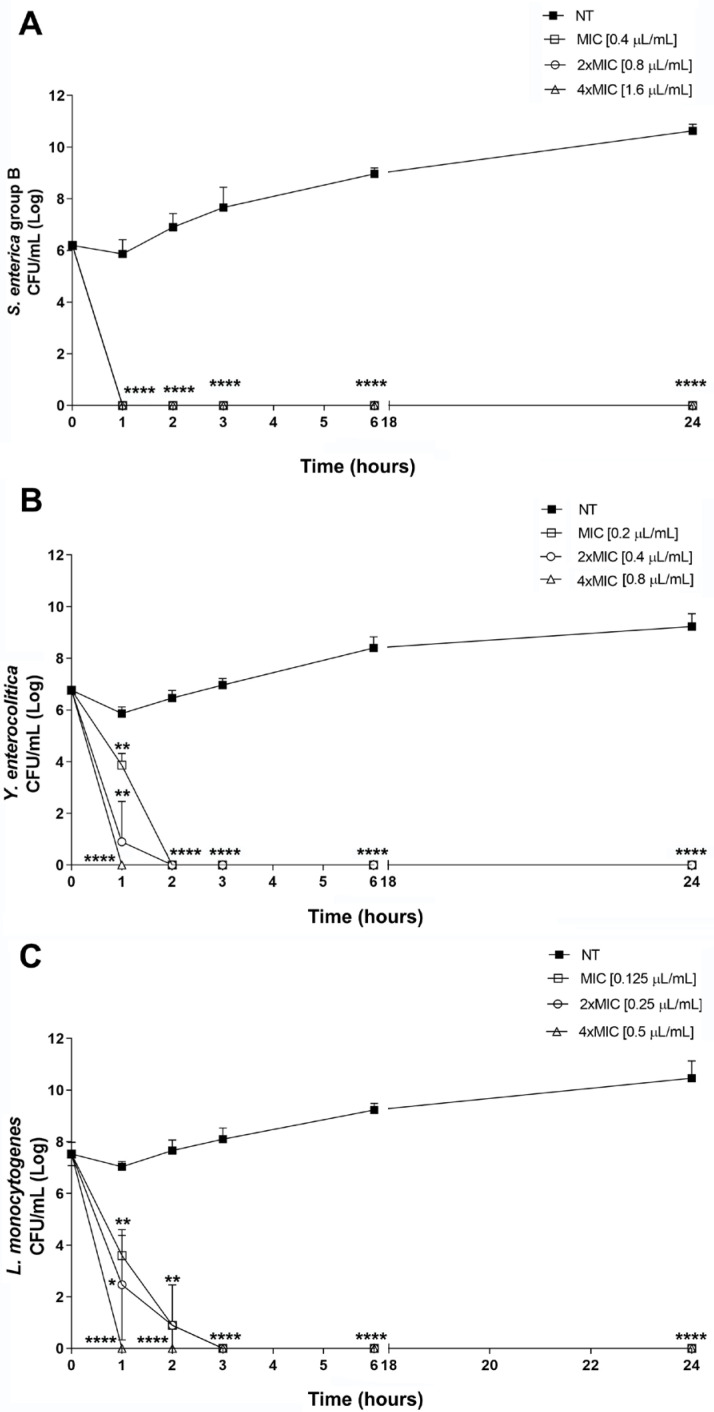
Time-kill assay of Thy-EO against foodborne bacteria. Time-kill curve of Thy-EO at different concentrations (MIC, 2×MIC, 4×MIC) against *S. enterica* (**A**), *Y. enterocolitica* (**B**), and *L. monocytogenes* (**C**). NT denotes untreated control. Statistical significance was assessed using Student’s *t*-test. Data represent the mean values of three independent experiments, with asterisks indicating statistically significant differences (**** *p* < 0.0001; ** *p* < 0.01; * *p* < 0.05).

**Figure 4 microorganisms-13-00037-f004:**
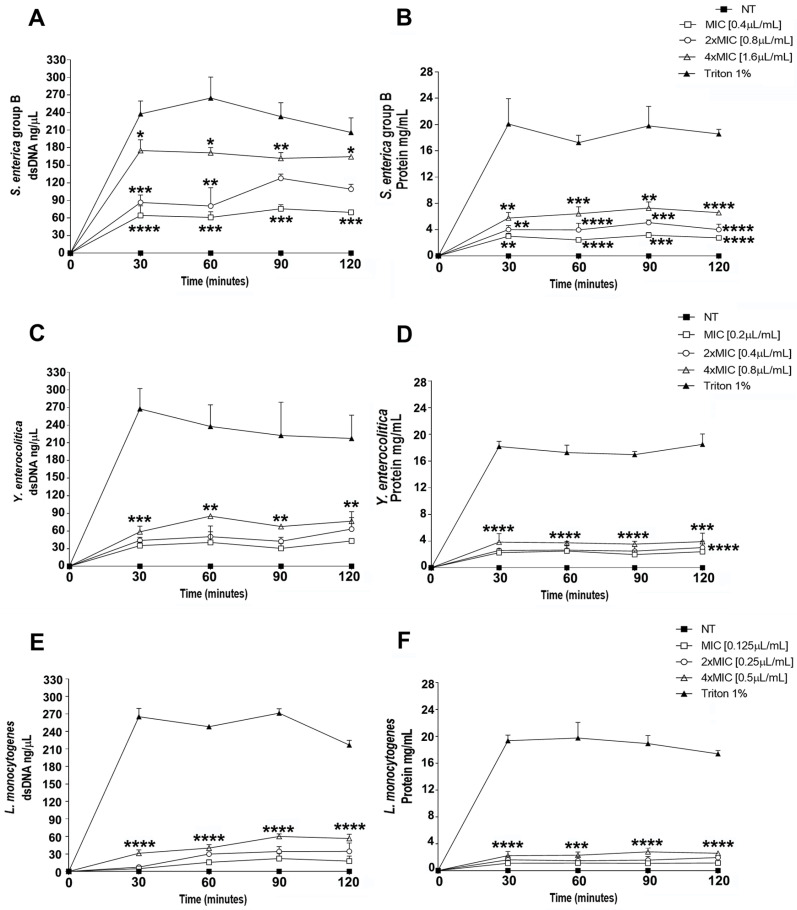
Effect of Thy-EO on bacterial cell membrane integrity. dsDNA (ng/μL) and protein (mg/mL) leakage induced by increasing concentrations of Thy-EO (MIC, 2×MIC, 4×MIC) against *S. enterica* (**A**,**B**), *Y. enterocolitica* (**C**,**D**), and *L. monocytogenes* (**E**,**F**). NT indicates untreated control, and Triton X-100 1% was used as positive control. The results are expressed as the absorbance of the sample (treated)—the absorbance of the negative control (NT). Data are expressed as the mean ± SD of experiments performed in triplicate. Student’s *t*-test was performed to evaluate statistical significance, and asterisks indicate statistically significant values (**** *p* < 0.0001; *** *p* < 0.001; ** *p* < 0.01; * *p* < 0.05).

**Table 1 microorganisms-13-00037-t001:** Antimicrobial activities of Thy-EO against enteric bacteria clinical isolates.

	Antimicrobial Agent			
	Thy-EO (µL/mL)		Gentamicin (μg/mL)	
	^a^ MIC	^b^ MBC	MIC	MBC
*S. enterica* group B	0.4	0.4/0.5	40	40
*Y. enterocolitica*	0.2	0.4/0.5	10	20
*L. monocytogenes*	0.125	0.2/0.3	10	10

^a^ MIC—minimum inhibitory concentration values are expressed as concentration μL/mL of Thy-EO or as concentration μg/mL of antibiotics. ^b^ MBC—minimum bactericidal concentration values are expressed as concentration μL/mL of Thy-EO or as concentration μg/mL of antibiotics.

## Data Availability

The original contributions presented in this study are included in the article/Appendix A. Further inquiries can be directed to the corresponding authors.

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
