# Peer review of "Thyme Essential Oil as a Potential Tool Against Common and Re-Emerging Foodborne Pathogens: Biocidal Effect on Bacterial Membrane Permeability"

_microorganisms, 2024, doi:10.3390/microorganisms13010037_

Round 1
Reviewer 1 Report
Comments and Suggestions for Authors
Rosario et al. investigated the potential of thyme essential oil (Thy-EO) as a natural antimicrobial agent against the most common and re-emerging foodborne bacteria, including S. enterica, Yersinia enterocolitica, and L. monocytogenes. The manuscript is very interesting and provides alternatives to antimicrobials since, over the past decade, foodborne diseases have become a significant public health concern, affecting millions of people globally. Additionally, the emergence of MDR is another concern. The manuscript is well-written, but I have some comments
1) The contents of bioactive substances vary significantly according to the products, plant, harvesting time, and even extraction and formulation methods. Please provide the analysis of the used product.
2) Please discuss the limitations of the use of phytogenic substances and the authors' recommendations.
3) Please discuss the potential effect mechanisms of Thymol as an antibacterial agent
4) I have marked several minor comments; please see the attached file. The name of the bacterial species must be written in italics. Also, please update the introduction with the most recent references.

The English could be improved to more clearly express the research.
Reviewer 2 Report
Comments and Suggestions for Authors
The article entitled "Thyme Essential Oil as a Potential Tool against Common and Re-Emerging Foodborne Pathogens: Biocidal Effect on Bacterial Membrane Permeability" investigates the pottential effects of thyme essential oils as natural antimicrobial agents against some well-known food pathogens such as Salmonella enterica, Yersinia enterocolitica and Listeria monocytogenes.
The article is very well-written and offers a fresh perspective on a subject that is of great interest. The introduction offers enough background information to help the reader have an overview of the research subject. The Materials and methods section accurately describes each method in an easy to understand and reproduce manner. The results are clearly presented and supported by graphs. The conclusion summarizes the results very well.
I reccommend this article to be accepted for publishing after minor English language editing.
Comments on the Quality of English LanguagePlease revise the English language (ex. Line 113 - instead of "at the dark" please use "in the dark").
Reviewer 3 Report
Comments and Suggestions for Authors
Review of the article: “Thyme Essential Oil as a Potential Tool against Common and Re-Emerging Foodborne Pathogens: Biocidal Effect on Bacterial Membrane Permeability”
Submission ID - microorganisms-3336617
The manuscript is generally interesting and well written. The authors have evaluated antimicrobial potential of Thyme essential oil against selected foodborn pathogens, namely: S. enterica, Yersinia enterocolitica, and Listeria monocytogenes.
In my opinion the most important weakness of the manuscript is the fact that antimicrobial activity of Thyme essential oil is well known and well documented. Thus, considering novelty the score is rather low. However, some parts of the study provide some results that are rarely presented by other authors e.g. determination of activity of volatile compounds.
Below I have presented detailed comments for authors. I would be grateful if the authors could consider my suggestions preparing final version of the manuscript.
1. Abstract – in my opinion the most important results (e.g. MIC, MBC values) should be presented in the abstract. In the current version of the abstract the authors only presented general idea of the study.
2. Methodology
Important weakness – only one sample of Thyme essential oils was used for the study. It is well known that composition of natural products (including essential oils) is importantly affected by many factors. Thus in my opinion it would be much more interesting if the authors have investigated and compared activity of several products delivered by different producers.
Bacterial strains - I do not understand why the authors used clinical isolates. It would be much better to use strains that are available in some collections (e.g. ATCC or DSM).
Membrane Permeability Assay - the authors evaluated the disruption of cell membrane integrity by determination of protein content in the medium after cells centrifugation. But the cells were grown in TS broth that contain large concentration of peptides. Thus, in my opinion only leakage of DNA should be considered.
3. Results are generally interesting and well presented. I have only two critical comments:
As I have written above, in my opinion determination of proteins should not been used for analysis of cell membrane disruption.
The quality of Figures 3 and 4 must be improved. Currently it is difficult to read the values presented in these figures.
4. Discussion
Thyme essential oils characterize with specific smell. I would be grateful for the comment if the concentration that is necessary for elimination of pathogenic bacteria can affect the smell (and taste) of food products.
Final decision – major revision.
Round 2
Reviewer 3 Report
Comments and Suggestions for Authors
The authors have addressed all my comments and suggestions. The manuscript can be accepted.
Author Response
Dear Reviewer 3,
in attachment the last version of the manuscript with required changes.
Best Regards,
Prof. Mariateresa Vitiello
